# Coping strategies adapted by Ghanaians during the COVID-19 crisis and lockdown: A population-based study

**Samuel Iddi**[1]*, **Dorcas Obiri-Yeboah**[2,3☯], **Irene Korkoi Aboh**[4], **Reginald Quansah**[5], **Samuel Asiedu Owusu**[3], **Nancy Innocentia Ebu Enyan**[4], **Ruby Victoria Kodom**[6], **Epaphrodite Nsabimana**[7], **Stefan Jansen**[7], **Benard Ekumah**[8], **Sheila A. Boamah**[9], **Godfred Odei Boateng**[10], **David Teye Doku**[3,11☯], **Frederick Ato Armah**[3,8]

1 Department of Statistics and Actuarial Science, University of Ghana, Legon-Accra, Ghana, 2 Department of Microbiology & Immunology, School of Medical Sciences, University of Cape Coast, Cape Coast, Ghana, 3 Directorate of Research, Innovation and Consultancy, University of Cape Coast, Cape Coast, Ghana, 4 Department of Adult Health, School of Nursing and Midwifery, University of Cape Coast, Cape Coast, Ghana, 5 Biological, Environmental & Occupational Health Sciences, School of Public Health, University of Ghana, Legon-Accra, Ghana, 6 Department of Public Administration and Health Services, University of Ghana, Legon-Accra, Ghana, 7 Mental Health & Behaviour Research Group, College of Medicine and Health Sciences, University of Rwanda, Kigali, Rwanda, 8 Department of Environmental Science, University of Cape Coast, Cape Coast, Ghana, 9 School of Nursing, Faculty of Health Sciences, McMaster University, Hamilton, Canada, 10 Global & Environmental Health Lab, Department of Kinesiology, College of Nursing and Health Innovation, University of Texas at Arlington, Arlington, Texas, United States of America, 11 Department of Population and Health, University of Cape Coast, Cape Coast, Ghana

☯ These authors contributed equally to this work.
* siddi@ug.edu.gh

## Abstract

### Background

The COVID-19 pandemic and control measures adopted by countries globally can lead to stress and anxiety. Investigating the coping strategies to this unprecedented crisis is essential to guide mental health intervention and public health policy. This study examined how people are coping with the COVID-19 crisis in Ghana and identify factors influencing it.

### Methods

This study was part of a multinational online cross-sectional survey on Personal and Family Coping with COVID-19 in the Global South. The study population included adults, ≥18 years and residents in Ghana. Respondents were recruited through different platforms, including social media and phone calls. The questionnaire was composed of different psychometrically validated instruments with coping as the outcome variable measured on the ordinal scale with 3 levels, namely, Not well or worse, Neutral, and Well or better. An ordinal logistic regression model using proportional odds assumption was then applied.

### Results

A total of 811 responses were included in the analysis with 45.2% describing their coping level as well/better, 42.4% as neither worse nor better and 12.4% as worse/not well. Many

**Data Availability Statement:** The dataset used in this paper is provided as a Supporting information file (see Appendix 2: "S2 Dataset.xlsx").

**Funding:** The author(s) received no specific funding for this work.

**Competing interests:** The authors have declared that no competing interests exist.

respondents (46.9%) were between 25–34 years, 50.1% were males while 79.2% lived in urban Ghana. Having pre-existing conditions increased the chances of not coping well (aOR = 1.86, 95%CI: 1.15–3.01). Not being concerned about supporting the family financially (aOR = 1.67, 95%CI: 1.06–2.68) or having the feeling that life is better during the pandemic (aOR = 2.37, 95%CI: 1.26–4.62) increased chances of coping well. Praying (aOR: 0.62, 95%CI: 0.43–0.90) or sleeping (aOR: 0.55, 95%CI: 0.34–0.89) more during the pandemic than before reduces coping.

## Conclusion

In Ghana, during the COVID-19 pandemic, financial security and optimism about the disease increase one's chances of coping well while having pre-existing medical conditions, praying and sleeping more during the pandemic than before reduces one's chances of coping well. These findings should be considered in planning mental health and public health intervention/policy.

## Introduction

The COVID-19 pandemic had and continues to impact severely on every aspect of what has been known as the 'normal' life. The pandemic has led to disruptions in daily life, social interactions, education, health, livelihood/employment, food security, safety and nutrition, politics, and economic activity. Governments around the world have responded differently to this pandemic and have achieved varying levels of success. The pandemic and the control measures instituted by governments resulted in fear of getting infected, dying, or losing a close friend or family member, psychological problems, and social panic [1–3].

In Ghana, as part of Government control measures, various forms of restrictions such as lockdown, and closure of schools and education institutions were implemented. These measures impacted academic, social, and economic activities [4]. The uncertainty associated with this unknown health crisis, the anxiety of sheltering-in-place, the realities of many parents working from their homes while at the same time home-schooling their children, and trying to meet their own family needs can create psychological stress. These life changes were all very sudden with very little time to plan or prepare for the impact which has created deleterious health outcomes effects with no clear end in sight. It is recognized that sudden events that disrupt routines and cause uncertainty can have a serious impact on the psychological wellbeing of people [5, 6]. Hence the World Health Organization (WHO) and other international agencies have recognized the need to include mental health interventions as part of efforts to support people through this crisis [7–9].

While countries and communities have employed different approaches in coping with the pandemic and lockdown, most households and individuals have had to employ idiosyncratic approaches in dealing with their peculiar challenges. At the individual level, the coping strategies have been influenced by characteristics such as gender, pre-existing health conditions, type of employment, and other socio-demographic factors [10–12]. Earlier studies also suggest that the utility of coping strategies are context-specific [13, 14]. Understanding what coping mechanism work in a given setting is therefore critical for the planning of interventions and public health policy in crisis such as the ongoing COVID-19 pandemic. However, as far as we know no study has investigated the coping strategies being adopted in response to the pandemic nor the factors associated with such strategies. Therefore, this study aims to examine the COVID-19 related coping mechanisms and associated factors among Ghanaians.

## Materials and methods

### Study design, study participants, and sample size

This study is part of a multi-country online cross-sectional survey on Personal and Family Coping with COVID-19 in the Global South. The larger study was designed to represent populations in 10 countries (Uganda, Bangladesh, Rwanda, Indonesia, Ghana, Brazil, Myanmar (Burma), Malaysia, Cameroon, and Kenya). The study population included participants aged 18 years or older, the ability to provide informed consent, and residents in partner countries. This paper utilizes data obtained from Ghana. Participants residing in Ghana were recruited from different social media platforms and personal contact via phone calls. The online questionnaire had in place measures to indicate if a participant is participating the first, second, etc. time. The analysis as used here is based on only the first responses. The sample size was estimated based on the nature of the analysis to be performed. A priori power analysis was performed using G*Power3.1 [15]. A total sample of 199 participants was required to achieve a generally accepted minimum level of power of 80 while detecting the smallest effect size (Cohen's $d$ = 0.2). For multivariate analysis, a sample size of 470 participants sufficed to perform any robust multivariate analysis. A large sample size to obtain more reliable results with greater precision and power was ensued due to less cost, time, and money because it was online. The study used an online survey design with respondents self-selecting to be part of the study. The questionnaire and psycho-educational feedback materials were designed to fit with the Global South context or approach. In this study, data on 813 respondents from the Ghana survey were analysed.

### Data collection and ethical clearance

In Ghana, participants were invited to take part in the study through different platforms, including social media (e.g. WhatsApp groups, email lists, Facebook, Twitter, and websites) and personal contact via phone calls and word of mouth information. The online questionnaire was generally composed of nine sections: section one collected data on socio-demographic information, section two collected information regarding COVID-19 and how respondents coped with it in a 'Before' and 'Since COVID-19' manner, while the other sections collected data not used in this present article. A copy of the questionnaire can be found in S1 Appendix. Participation was voluntary and participants could skip questions they did not wish to respond to. The online form required the participant to read the background information on the study and then select to indicate if they were ready to participate before they were able to proceed to respond to the questions. Also, participants received instant feedback on their responses which also gave them some tailored advice on the management and control of COVID-19 infection. The language for the study was English. Data collection started on 13[th] July and continued until the end of September 2020. Approval for the study was sought from the University of Cape Coast Institutional Review Board (UCCIRB/EXT/2020/12). The dataset used in this paper is provided as a supporting information file (see S1 Dataset).

### Data analysis

We summarized each socio-demographic variable, COVID-19 characteristics, and engagement in various activities using STATA version 14 and presented them as frequencies. In addition, bivariate associations of these characteristics were tested using the Chi-square test. The outcome variable was coping which was measured on the ordinal scale with 3 levels, namely, Not well or worse, Neutral, and Well or better. We then applied the ordinal logistic regression model using the proportional odds assumption. The proportional odds assumption was

verified by performing a likelihood ratio test between a general multinomial regression model and the proportional odds model.

To conduct the data analysis, first, we considered crude models, where we fitted separate models with only socio-demographic factors or COVID-19 related characteristics or level of engagement that were significant at 0.1% confidence interval (alpha = 0.10) in the bivariate analysis. For the socio-demographic factors, age and sex were included although they were not significant because of their potential as confounders. Next, we considered separate models for COVID-19 related characteristics and level of engagement while controlling for the demographic factors and an adjusted model when both sets of factors were included in the model.

## Results

### Characteristics of the study population

A total of 811 participant responses were used in the analysis. The characteristics of study participants are shown in Table 1. The highest proportion of the respondents (46.9%) were between 25–34 years, 50.1% were males, 79.2% lived in urban Ghana, (61.9%) were in a relationship of some sort, (47.9%) had between 1–3 children, (32.4%) had at least a bachelor degree, and few were employed in the non-governmental sector (11.2%) or belong to high-income economic category (2.0%).

### COVID-19 related characteristics of participants

In terms of COVID-19 related characteristics (Table 2), 2.3% of the participants indicated that they have been infected with the virus, 2.8% had a household member who has been infected and 17.9% had someone close to them (a relative or friend) has been infected with the virus. Also, 2.2% of the participants had someone close dying from the infection, the majority (77.4%) were extremely concerned about their health and that of a family member. Eleven percent had pre-existing medical conditions and 77.8% were concerned about their family's finances.

### Engagement in various activities 'during' compared with' before' the COVID-19 pandemic

In the majority of the participants, the main activities engaged in were television viewing (45.5%), their engagement in income-generating activities from home (35%), performance of household chores (52.8%), engagement in sports (40.2%), and devotion to prayers (48.6%), and quality of sleep (49.7%) have not seen any changes. But 45.4% and 46.4% spent more time on social media and talking on the phone during the COVID-19 outbreak than before (Table 3).

### Levels of coping by individual participants

Of 805 individuals who answered questions on coping strategy, representing 99.3% of eligible participants, 45.2% described their coping strategy as well or better, 42.4% described theirs as neither worse nor better and 12.4% describing theirs as worse or not well (Fig 1).

### Association of socio-demographic, COVID-19 related characteristics, engagement of activities and coping

Of the demographic characteristics, education (p = 0.005), and economic class (p<0.0001) were significantly associated with coping. The proportion of participants who claimed to have

**Table 1. Socio-demographic characteristics of participants (N = 811).**

| Variable | Frequency (n) | Percentage (%) |
|---|---|---|
| **Age (years)** | | |
| 18–24 | 51 | 6.3 |
| 25–34 | 379 | 46.9 |
| 35–44 | 244 | 30.2 |
| 45–54 | 96 | 11.9 |
| 55–64 | 26 | 3.6 |
| 65+ | 9 | 1.1 |
| Missing data | 3 | |
| **Gender** | | |
| Female | 404 | 49.9 |
| Male | 406 | 50.1 |
| Missing data | 1 | |
| **Residence** | | |
| Urban | 623 | 79.2 |
| Rural | 164 | 20.8 |
| Missing data | 24 | |
| **Relationship Status** | | |
| *In a relationship | 492 | 61.9 |
| Not in a relationship | 303 | 38.1 |
| Missing data | 16 | |
| **Number of Children** | | |
| 0 | 353 | 44.1 |
| 1–3 | 384 | 47.9 |
| 4+ | 64 | 8.0 |
| Missing data | 10 | |
| **Level of Education** | | |
| Secondary or lower | 26 | 3.2 |
| Post-secondary | 189 | 23.4 |
| Bachelor's | 261 | 32.4 |
| Masters | 237 | 29.4 |
| Doctorate | 93 | 11.5 |
| Missing data | 5 | |
| **Employment** | | |
| Unemployed | 129 | 16.0 |
| Non-government work | 95 | 11.8 |
| Government work | 580 | 72.1 |
| Missing data | 7 | |
| **Economic category** | | |
| Low income | 120 | 14.9 |
| Lower middle income | 483 | 59.8 |
| Higher middle income | 189 | 23.4 |
| High income | 16 | 2.0 |
| Missing data | 3 | |

*All who indicated being married, cohabiting, and having a partner were considered as being in a relationship. Those who were single, widowed, or divorced were considered as not being in a relationship.

**Table 2. COVID-19 related characteristics of participants (N = 811).**

| Variable | Frequency (n) | Percentage (%) |
|---|---|---|
| **COVID-19 Infection** | | |
| Yes | 19 | 2.3 |
| No | 721 | 89.0 |
| Not sure | 70 | 8.6 |
| Missing data | 1 | |
| **Household Infected** | | |
| Yes | 23 | 2.8 |
| No | 734 | 90.7 |
| Not sure | 52 | 6.4 |
| Missing data | 2 | |
| **Someone close Infected** | | |
| Yes | 145 | 17.9 |
| No | 602 | 74.4 |
| Not sure | 62 | 7.7 |
| Missing data | 2 | |
| **Someone close died from the COVID-19** | | |
| Yes | 18 | 2.2 |
| No | 764 | 94.4 |
| Not sure | 27 | 3.3 |
| Missing data | 2 | |
| **Concerned about own/family health** | | |
| Not at all concerned | 3 | 0.4 |
| Slightly concerned | 23 | 2.8 |
| Somewhat concerned | 43 | 5.3 |
| Moderately concerned | 114 | 14.1 |
| Extremely concerned | 627 | 77.4 |
| Missing data | 1 | |
| **Pre-existing condition** | | |
| Yes | 89 | 11.0 |
| No | 657 | 81.2 |
| Not sure | 63 | 7.8 |
| Missing data | 2 | |
| **Concerned about supporting your family financially** | | |
| Yes | 629 | 77.8 |
| No | 130 | 16.1 |
| Not sure | 50 | 6.2 |
| Missing data | 2 | |
| **Difficult to switch off media reporting on COVID-19** | | |
| Easy | 174 | 21.5 |
| Very easy | 154 | 19.1 |
| Neither easy/difficult | 275 | 33.91 |
| Difficult | 144 | 17.8 |
| Very difficult | 61 | 7.5 |
| Missing Data | 3 | |
| **Better or worse life** | | |
| Same | 411 | 50.7 |
| Better | 73 | 9.0 |

(*Continued*)

**Table 2.** (Continued)

| Variable | Frequency (n) | Percentage (%) |
|---|---|---|
| Much better | 10 | 1.2 |
| Worse | 270 | 33.3 |
| Much worse | 46 | 5.7 |
| Missing Data | 1 | |
| **Country control** | | |
| Neutral | 161 | 19.9 |
| Very well controlled | 66 | 8.1 |
| Somehow controlled | 317 | 39.1 |
| Not very well controlled | 222 | 27.4 |
| Not well controlled at all | 44 | 5.4 |
| Missing data | 1 | |

coped better during COVID-19 infection generally increased with increasing levels of education, Thirty-seven percent of those with post-secondary education claimed to have coped better. The correspondent proportion with those with bachelor's degree, masters' and doctorate degree were 44.4%, 46.4%, and 59.3%, respectively. However, with respect to the economic category, there was no specific order (Table 4).

With respect to COVID-19 related characteristics, infection of a household member (p = 0.013), infection of someone close to participants (p = 0.031), participants concerned of own health and the health of a family member (p = 0.001), concern about supporting the family financially (p = 0.010), having difficulties switching off from media reporting on COVID-19 (p<0.001), life becoming better or worse since the COVID-19 crisis started (p<0.001), and control of COVID-19 by the government (p<0.001) were significantly associated with coping. Praying (p<0.001), resting or relaxing (p<0.001), and sleeping (p<0.001) were all significantly associated with coping (Table 5).

We observe from Table 6 that all the variables related to engagement in various activities 'during' and 'before' COVID-19 pandemic including watching Television (p<0.001), time spent on the internet not related to work (p = 0.002), time spent on social media not related to work (p = 0.013), working for income from home (p = 0.036), performing household chores

**Table 3.** Levels of engagement in various activities 'during' compared with 'before' the COVID-19 pandemic by participants (N = 811).

| Variable | Frequency (n) | Percentage (%) |
|---|---|---|
| **Watching Television** | | |
| Same as before | 369 | 45.5 |
| Less than before | 158 | 19.5 |
| More than before | 245 | 30.2 |
| Prefer not to say | 39 | 4.8 |
| **Time on internet (not for work)** | | |
| Same as before | 329 | 40.8 |
| Less than before | 98 | 12.2 |
| More than before | 366 | 45.4 |
| Prefer not to say | 13 | 1.6 |
| Missing data | 5 | |
| **Time on social media (not for work)** | | |

(*Continued*)

**Table 3.** (Continued)

| Variable | Frequency (n) | Percentage (%) |
|---|---|---|
| Same as before | 356 | 44.0 |
| Less than before | 108 | 13.3 |
| More than before | 328 | 40.5 |
| Prefer not to say | 17 | 2.1 |
| Missing data | 2 | |
| **Working for income from home** | | |
| Same as before | 282 | 35.0 |
| Less than before | 182 | 22.6 |
| More than before | 252 | 31.3 |
| Prefer not to say/missing data | 90 | 11.2 |
| Missing data | 5 | |
| **Performing household chores** | | |
| Same as before | 437 | 52.8 |
| Less than before | 75 | 9.3 |
| More than before | 276 | 34.1 |
| Prefer not to say | 31 | 3.8 |
| Missing data | 2 | |
| **Engaging in sports** | | |
| Same as before | 325 | 40.2 |
| Less than before | 287 | 35.5 |
| More than before | 124 | 15.3 |
| Prefer not to say | 72 | 8.9 |
| Missing data | 3 | |
| **Talking on phone** | | |
| Same as before | 347 | 43.1 |
| Less than before | 78 | 9.7 |
| More than before | 374 | 46.4 |
| Prefer not to say | 7 | 0.9 |
| Missing data | 5 | |
| **Praying** | | |
| Same as before | 393 | 48.6 |
| Less than before | 89 | 11.0 |
| More than before | 304 | 37.6 |
| Prefer not to say | 23 | 2.8 |
| Missing data | 2 | |
| **Resting/relaxing** | | |
| Same as before | 326 | 40.3 |
| Less than before | 155 | 19.2 |
| More than before | 324 | 40.0 |
| Prefer not to say/missing data | 4 | 0.5 |
| Missing data | 2 | |
| **Sleeping** | | |
| Same as before | 401 | 49.7 |
| Less than before | 163 | 20.2 |
| More than before | 237 | 29.4 |
| Prefer not to say | 6 | 0.7 |
| Missing data | 4 | |

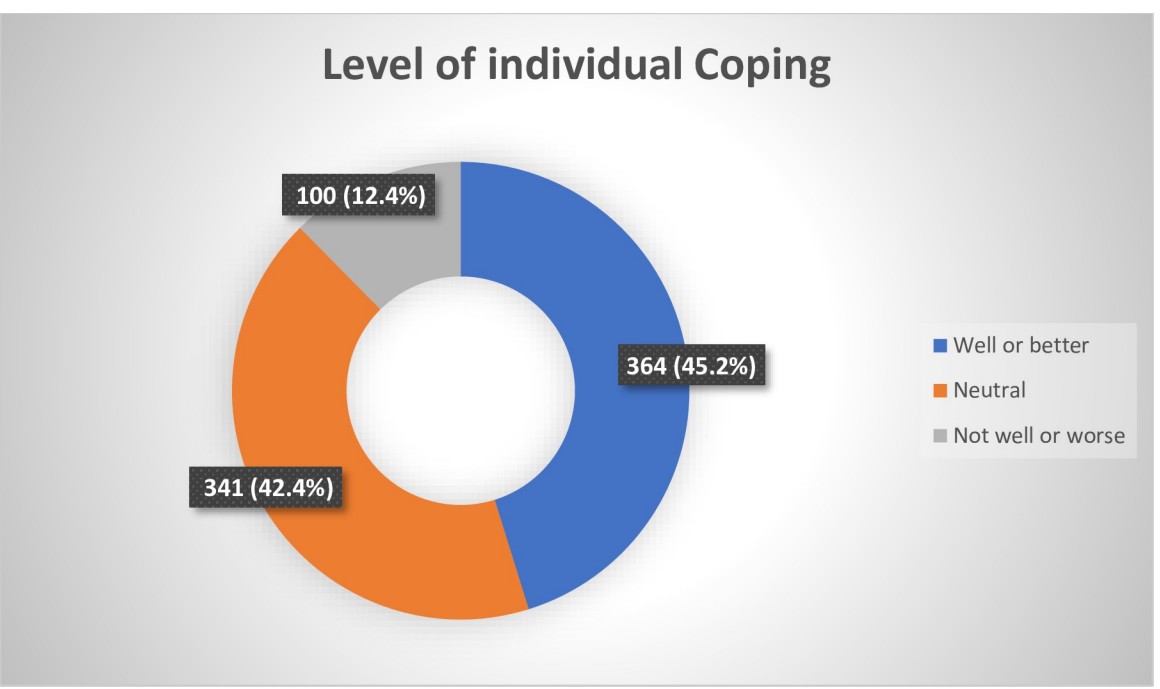

**Fig 1. Level of coping by individual participants (N = 805).**

(p<0.001), engagement in sport (p = 0.005), talking on the phone (p<0.001) were associated with coping.

The results of the ordered logistic regression are shown in Table 7. Note that the model was fitted using the R statistical software which uses a negative parameterization of the coefficient for the fixed effect similar to what is used in STATA. Thus, the interpretation of the odds does not take the form of an ordinary logistic regression. We also verified the proportional odds assumption for the adjusted model by performing a likelihood ratio test that compared the proportional odds model with the general multinomial regression model while keeping socio-demographic variables and significant COVID- 19 related characteristics, and level of engagement in activities variables. We obtained a non-significant p-value of 0.3668 ($X^2$ = 42.43, $df$ = 42) and thus concluded that the proportional odds assumption is plausible. Among participants who are 18–24 years, the odds of not coping well (i.e. not coping well or worse vs. neutral or better) is 65%, 61%, and 70% lower compared to those in the age group 25–34, 35–44, and 45 54 years, respectively after adjusting for potential cofounders. The odds of not coping well is 58%, 98%, and 236% increase in those of the low-income group compared to those in lower middle income, high middle income, and high-income groups, respectively but the lower limit of some of the associations included unity.

Having pre-existing conditions increased participants' chances of not coping well compared to when participants had no pre-existing condition (aOR = 2.12, 95%CI: 1.27–3.55). Not being concerned about supporting family finances (aOR = 1.67, 95%CI: 1.06–2.68) or having the feeling that life is better during the pandemic (aOR = 2.37, 95%CI: 1.26–4.62) increases participants' chances of coping well. The chances of coping well increased when participants perceived that Ghana had very well controlled the infection (aOR: 2.75, 95%CI: 1.32–5.88) or somehow been able to control (aOR = 1.78, 95%CI: 1.15–2.76) the pandemic compared to when they are neutral about it. Praying more than before (aOR: 0.62, 95%CI: 0.43–0.90) or sleeping more than before (aOR: 0.55, 95%CI: 0.34–0.89) reduces one's chances of coping well (Table 7).

**Table 4. Association between coping and demographic characteristics.**

| Variable | Coping | | | |
|---|---|---|---|---|
| | Well or better, N = 364 | Neutral, N = 341 | Not well or worse, N = 100 | P-value |
| **Age (years)** | n (%) | n (%) | n (%) | 0.288 |
| 18–24 | 27 (52.9) | 19 (37.3) | 5 (9.8) | |
| 25–34 | 166 (43.9) | 162 (42.9) | 50 (13.2) | |
| 35–44 | 106 (44.2) | 111 (46.2) | 23 (9.6) | |
| 45–54 | 42 (43.8) | 37 (38.5) | 17 (17.7) | |
| 55–64 | 15 (53.6) | 9 (32.1) | 4 (14.3) | |
| 65+ | 7 (77.8) | 2 (22.2) | 0 (0.0) | |
| Missing data | 1 | 1 | 1 | |
| **Gender** | | | | 0.200 |
| Female | 168 (42.1) | 179 (44.9) | 52 (13.0) | |
| Male | 196 (48.4) | 161 (39.8) | 48 (11.9) | |
| Missing data | 0 | 1 | 0 | |
| **Residence** | | | | 0.210 |
| Urban | 287 (46.4) | 263 (42.6) | 68 (16.0) | |
| Rural | 69 (42.3) | 68 (41.7) | 26 (16.0) | |
| Missing data | 8 | 10 | 6 | |
| **Relationship Status** | | | | 0.474 |
| In relationship | 229 (47.0) | 198 (40.7) | 60 (12.3) | |
| Not in relationship | 129 (42.7) | 135 (44.7) | 38 (12.6) | |
| Missing data | 6 | 8 | 2 | |
| **Number of Children** | | | | 0.984 |
| 0 | 161 (45.7) | 147 (41.8) | 44 (12.5) | |
| 1–3 | 169 (44.6) | 162 (42.7) | 48 (12.7) | |
| 4+ | 31 (48.4) | 25 (39.1) | 8 (12.5) | |
| Missing data | 3 | 7 | 0 | |
| **Level of Education** | | | | *0.005* |
| Secondary or lower | 14 (53.8) | 7 (26.9) | 5 (19.2) | |
| Post-secondary | 69 (36.9) | 85 (45.5) | 33 (17.6) | |
| Bachelor's | 115 (44.4) | 109 (42.1) | 35 (13.5) | |
| Masters | 110 (46.4) | 107 (45.1) | 20 (8.4) | |
| Doctorate | 54 (59.3) | 31 (34.1) | 6 (6.6) | |
| Missing data | 2 | 2 | 1 | |
| **Employment** | | | | 0.350 |
| Unemployed | 52 (40.3) | 57 (44.2) | 20 (15.5) | |
| Non-government work | 46 (48.4) | 34 (35.8) | 15 (15.8) | |
| Government work | 265 (46.2) | 244 (42.5) | 65 (11.3) | |
| Missing data | 1 | 6 | 0 | |
| **Economic category** | | | | *<0.0001* |
| Low income | 43 (35.8) | 51 (42.5) | 26 (21.7) | |
| Lower middle income | 209 (43.7) | 218 (45.6) | 51 (10.7) | |
| Higher middle income | 99 (52.4) | 71 (37.6) | 19 (10.1) | |
| High income | 11 (73.3) | 1 (6.7) | 3 (20.0) | |
| Missing data | 2 | 0 | 1 | |

**Table 5. Association between coping and COVID-19 related characteristics of participants.**

| Variable | Coping | | | |
|---|---|---|---|---|
| | Well or better, N = 364 | Neutral, N = 341 | Not well or worse, N = 100 | P- value |
| **COVID-19 Infection** | n (%) | n (%) | n (%) | 0.805 |
| Yes | 10 (2.7) | 7 (2.1) | 2 (2.0) | |
| No | 326 (89.6) | 299 (87.9) | 90 (90.0) | |
| Not sure | 28 (7.7) | 34 (10.0) | 8 (8.0) | |
| Missing data | 0 | 1 | 0 | |
| **Household Infected** | | | | *0.013* |
| Yes | 5 (1.4) | 14 (4.1) | 4 (4.0) | |
| No | 342 (94.2) | 295 (86.8) | 91 (91.0) | |
| Not sure | 16 (4.4) | 31 (9.1) | 5 (5.0) | |
| Missing data | 1 | 1 | 0 | |
| **Someone close Infected** | | | | *0.031* |
| Yes | 53 (14.6) | 63 (18.5) | 27 (27.3) | |
| No | 286 (78.8) | 248 (72.7) | 64 (64.6) | |
| Not sure | 24 (6.6) | 30 (8.8) | 8 (8.1) | |
| Missing data | 1 | 0 | 1 | |
| **Someone close died** | | | | 0.080 |
| Yes | 7 (1.9) | 8 (2.3) | 3 (3.0) | |
| No | 344 (95.0) | 325 (95.3) | 89 (89.0) | |
| Not sure | 11 (3.0) | 8 (2.3) | 8 (8.0) | |
| Missing data | 2 | 0 | 0 | |
| **Concerned about own/family health** | | | | *0.001* |
| Not at all concerned | 2 (0.6) | 0 (0.0) | 1 (1.0) | |
| Slightly concerned | 16 (4.4) | 3 (0.9) | 3 (3.0) | |
| Somewhat concerned | 12 (3.3) | 29 (8.5) | 2 (2.0) | |
| Moderately concerned | 49 (13.5) | 55 (16.1) | 9 (9.0) | |
| Extremely concerned | 284 (78.2) | 254 (74.5) | 85 (85.0) | |
| Missing data | 1 | 0 | 0 | |
| **Pre-existing condition** | | | | 0.097 |
| Yes | 30 (8.3) | 43 (12.6) | 16 (16.0) | |
| No | 308 (85.1) | 269 (78.9) | 75 (75.0) | |
| Not sure | 24 (6.6) | 29 (8.5) | 9 (9.0) | |
| Missing data | 2 | 0 | 0 | |
| **Concerned about supporting your family financially** | | | | *0.010* |
| Yes | 266 (73.3) | 270 (79.2) | 88 (88.9) | |
| No | 73 (20.1) | 50 (14.7) | 6 (6.1) | |
| Not sure | 24 (6.6) | 21 (6.2) | 5 (5.1) | |
| Missing data | 1 | 0 | 1 | |
| **Difficult to switch off media reporting on COVID-19** | | | | *<0.001* |
| Easy | 91 (25.1) | 63 (18.5) | 17 (17.0) | |
| Very easy | 91 (25.1) | 43 (12.6) | 18 (17.0) | |
| Neither easy/difficult | 120 (33.1) | 134 (39.4) | 21 (21.0) | |
| Difficult | 42 (11.6) | 71 (20.9) | 31 (31.0) | |
| Very difficult | 18 (5.0) | 29 (8.5) | 13 (13.0) | |
| Missing Data | 2 | 1 | 0 | |
| **Better or worse life** | | | | *<0.001* |
| Same | 216 (59.5) | 171 (50.1) | 21 (21.0) | |

*(Continued)*

**Table 5.** (Continued)

| Variable | Coping | | | P- value |
|---|---|---|---|---|
| | **Well or better, N = 364** | **Neutral, N = 341** | **Not well or worse, N = 100** | |
| Better | 52 (14.3) | 18 (5.3) | 3 (3.0) | |
| Much better | 6 (1.7) | 3 (0.9) | 1 (1.0) | |
| Worse | 82 (22.6) | 132 (38.7) | 53 (53.0) | |
| Much worse | 7 (1.96) | 17 (5.0) | 22 (22.0) | |
| Missing Data | 1 | 0 | 0 | |
| **Country control** | | | | *<0.001* |
| Neutral | 61 (16.8) | 80 (23.5) | 20 (20.0) | |
| Very well controlled | 45 (12.4) | 14 (4.1) | 5 (5.0) | |
| Somehow controlled | 159 (43.8) | 129 (37.8) | 27 (27.0) | |
| Not very well controlled | 86 (23.7) | 100 (29.3) | 35 (35.0) | |
| Not well controlled at all | 12 (3.3) | 18 (5.3) | 13 (13.0) | |
| Missing data | 1 | 0 | 0 | |

## Discussions

This article examined the correlates of COVID-19 coping strategies in Ghana. We found that majority of the respondents were extremely concerned about their health and that of a family member during the pandemic. Eleven percent had pre-existing medical conditions and 77.8% were concerned about their family's finances as a result of COVID-19. Being between 25–34 or 35–44 or 45–54 years or praying more than before or sleeping more than before reduces one's chances of coping. However, being of higher middle-income status, having no pre-existing health conditions, or not being bothered about supporting a family member financially enhanced positive coping.

The COVID-19 pandemic has posed environmental, social, economic, and health threats to every country in the world. The measure such as social distancing, quarantine, isolation, and lockdown implemented by the government to contain the pandemic have not only affected the social and economic life of people but also created fear and panic in many populations. The fear and panic created by the pandemic compelled people to adopt various coping strategies

**Table 6. Association between coping and levels of engagement in various activities during and 'before' the COVID-19 pandemic by participants.**

| Variable | Coping | | | P- value |
|---|---|---|---|---|
| | **Well or better, N = 364** | **Neutral, N = 341** | **Not well or worse, N = 100** | |
| **Watching Television** | | | | *<0.001* |
| Same as before | 190 (52.2) | 159 (46.6) | 18 (18.0) | |
| Less than before | 56 (15.4) | 70 (20.5) | 32 (32.0) | |
| More than before | 93 (25.5) | 101 (29.6) | 47 (47.0) | |
| Prefer not to say | 25 (6.9) | 11 (3.2) | 3 (3.0) | |
| **Time on internet** | | | | *0.002* |
| Same as before | 156 (43.1) | 148 (43.8) | 23 (23.0) | |
| Less than before | 39 (10.8) | 39 (11.5) | 20 (20.0) | |
| More than before | 161 (44.5) | 144 (42.6) | 57 (57.0) | |
| Prefer not to say | 6 (1.7) | 7 (2.1) | 0 (0.0) | |
| Missing data | 2 | 3 | 0 | |
| **Time on social media** | | | | |
| Same as before | 173 (47.5) | 153 (45.1) | 28 (28.0) | *0.013* |

*(Continued)*

**Table 6.** (Continued)

| Variable | Coping | | | P- value |
|---|---|---|---|---|
| | Well or better, N = 364 | Neutral, N = 341 | Not well or worse, N = 100 | |
| Less than before | 46 (12.6) | 41 (12.1) | 21 (21.0) | |
| More than before | 135 (37.1) | 140 (41.3) | 49 (49.0) | |
| Prefer not to say | 10 (2.7) | 5 (1.5) | 2 (2.0) | |
| Missing data | 0 | 2 | 0 | |
| **Working for income from home** | | | | *0.036* |
| Same as before | 136 (37.6) | 120 (35.5) | 24 (24.0) | |
| Less than before | 72 (19.9) | 76 (22.5) | 33 (33.0) | |
| More than before | 119 (32.9) | 97 (28.7) | 33 (33.0) | |
| Prefer not to say | 35 (9.7) | 45 (13.3) | 10 (10.0) | |
| Missing data | 2 | 3 | 0 | |
| **Performing household chores** | | | | *<0.001* |
| Same as before | 224 (61.9) | 165 (48.4) | 35 (35.0) | |
| Less than before | 25 (6.9) | 32 (9.4) | 18 (18.0) | |
| More than before | 102 (28.2) | 131 (38.4) | 41 (41.0) | |
| Prefer not to say | 11 (3.0) | 13 (3.8) | 6 (6.0) | |
| Missing data | 2 | 0 | 0 | |
| **Engaging in sports** | | | | *0.005* |
| Same as before | 162 (44.8) | 137 (40.3) | 23 (23.0) | |
| Less than before | 119 (32.9) | 126 (37.1) | 40 (40.0) | |
| More than before | 50 (13.8) | 51 (15.0) | 23 (23.0) | |
| Prefer not to say | 31 (8.6) | 26 (7.6) | 14 (14.0) | |
| Missing data | 2 | 1 | 0 | |
| **Talking on phone** | | | | *<0.001* |
| Same as before | 179 (49.7) | 144 (42.4) | 23 (23.0) | |
| Less than before | 25 (6.9) | 34 (10.0) | 17 (17.0) | |
| More than before | 153 (42.5) | 159 (46.8) | 59 (59.0) | |
| Prefer not to say | 3 (0.8) | 3 (0.9) | 1 (1.0) | |
| Missing data | 4 | 1 | 0 | |
| **Praying** | | | | *<0.001* |
| Same as before | 203 (55.9) | 159 (46.8) | 30 (30.0) | |
| Less than before | 28 (7.7) | 39 (11.5) | 21 (21.0) | |
| More than before | 123 (33.9) | 134 (39.4) | 44 (44.0) | |
| Prefer not to say | 9 (2.5) | 8 (2.4) | 5 (5.0) | |
| Missing data | 1 | 1 | 0 | |
| **Resting/relaxing** | | | | *<0.001* |
| Same as before | 172 (47.4) | 132 (38.7) | 22 (22.2) | |
| Less than before | 48 (13.2) | 71 (20.8) | 35 (35.4) | |
| More than before | 142 (39.1) | 137 (40.2) | 40 (40.4) | |
| Prefer not to say | 1 (0.3) | 1 (0.3) | 2 (2.0) | |
| Missing data | **1** | **0** | **1** | |
| **Sleeping** | | | | *<0.001* |
| Same as before | 217 (59.9) | 154 (45.3) | 28 (28.3) | |
| Less than before | 49 (13.5) | 76 (22.4) | 37 (37.4) | |
| More than before | 94 (26.0) | 109 (32.1) | 31 (31.3) | |
| Prefer not to say | 2 (0.6) | 1 (0.3) | 3 (3.0) | |
| Missing data | 2 | 1 | 1 | |

**Table 7. Association of socio-demographic, COVID- 19 related characteristics, and engagement in activities with coping (n = 811).**

| Determinants of Interest | Coping | |
|---|---|---|
| | Crude OR (95%CI) | Adjusted OR (95% CI) |
| **Sociodemographic characteristics** | | |
| *Sex* | | |
| Female | 1.00 | 1.00 |
| Male | 1.21 (0.92–1.58) | 1.33 (0.96, 1.84) |
| *Age (years)* | | |
| 18–24 | 1.00 | 1.00 |
| 25–34 | **0.49 (0.26–0.89)** | **0.35 (0.17 0.71)** |
| 35–44 | **0.39 (0.20–0.75)** | **0.39 (0.18 0.84)** |
| 45–54 | **0.26 (0.12–0.56)** | **0.30 (0.12 0.72)** |
| 55–64 | 0.40 (0.15–1.08) | 0.42 (0.13 1.34) |
| 65+ | 1.83 (0.39–13.34) | 2.44 (0.35 24.94) |
| *Level of Education* | | |
| Secondary or lower | 1.00 | 1.00 |
| Post-secondary | 0.68 (0.28–1.63) | 0.51 (0.19 1.34) |
| Bachelor's | 0.91 (0.38–2.13) | 0.62 (0.23 1.62) |
| Masters | 1.17 (0.48–2.78) | 0.80 (0.29 2.14) |
| Doctorate | 1.84 (0.70–4.79) | 1.05 (0.35 3.14) |
| *Economic category* | | |
| Low income | 1.00 | 1.00 |
| Lower middle income | **1.75 (1.16–2.66)** | 1.58 (0.98 2.55) |
| Higher middle income | **2.16 (1.33–3.53)** | **1.98 (1.14 3.48)** |
| High income | **3.91 (1.18–15.46)** | 3.36 (0.87 15.62) |
| **COVID-19 related characteristics** | | |
| *Have being infected* | | |
| Yes | 1.00 | 1.00 |
| No | 1.75 (0.66–4.94) | 0.47 (0.16–1.33) |
| Not sure | 1.41 (0.48–4.35) | 0.51 (0.15–1.62) |
| *Household Infected* | | |
| Yes | 1.00 | 1.00 |
| No | 0.57 (0.23–1.38) | 2.05 (0.81–5.23) |
| Not sure | 0.77 (0.26–2.30) | 1.88 (0.59–6.06) |
| *Someone close Infected* | | |
| Yes | 1.00 | 1.00 |
| No | 0.75 (0.49–1.15) | 1.29 (0.81 2.06) |
| Not sure | 0.67 (0.33.1.34) | 1.51 (0.71 3.25) |
| *Someone close died from the COVID-19* | | |
| Yes | 1.00 | 1.00 |
| No | 1.24 (0.47–3.39) | 0.99 (0.32–2.97) |
| Not sure | 2.05 (0.57–7.38) | 0.50 (0.12–2.04) |
| *Concerned about own/family health* | | |
| Not at all concerned | 1.00 | 1.00 |
| Slightly concerned | 0.32 (0.02–8.32) | 2.44 (0.08 42.82) |
| Somewhat concerned | 1.28 (0.10–30.77) | 0.54 (0.02 8.00) |
| Moderately concerned | 0.60 (0.05–13.97) | 1.17 (0.05 16.61) |
| Extremely concerned | 0.53 (0.05–12.24) | 1.39 (0.06 19.15) |

(*Continued*)

**Table 7.** (Continued)

| Determinants of Interest | Coping | |
|---|---|---|
| | **Crude OR (95%CI)** | **Adjusted OR (95% CI)** |
| *Pre-existing condition* | | |
| Yes | 1.00 | 1.00 |
| No | 0.68 (0.43–1.07) | **2.12 (1.27 3.55)** |
| Not sure | 0.83 (0.42–1.61) | 1.36 (0.66 2.81) |
| *Concerned about supporting your family financially* | | |
| Yes | 1.00 | 1.00 |
| No | **0.49 (0.32–0.74)** | **1.67 (1.06 2.68)** |
| Not sure | 1.02 (0.56–1.84) | 0.97 (0.51 1.89) |
| *Difficult to switch off media reporting on COVID-19* | | |
| Easy | 1.00 | 1.00 |
| Very easy | 0.83 (0.51–1.33) | 1.10 (0.66 1.84) |
| Neither easy/difficult | 1.14 (0.77–1.71) | 0.91 (0.59 1.42) |
| Difficult | **2.23 (1.41–3.56)** | **0.44 (0.26 0.73)** |
| Very difficult | 1.67 (0.89–3.14) | 0.56 (0.28 1.13) |
| *Better or worse life* | | |
| Same | 1.00 | 1.00 |
| Better | **0.52 (0.29–0.92)** | **2.37 (1.26 4.62)** |
| Much better | 0.72 (0.16–2.65) | 2.51 (0.58 12.61) |
| Worse | **2.59 (1.89–3.58)** | **0.49 (0.34 0.70)** |
| Much worse | **7.74 (3.96–15.34)** | **0.13 (0.06 0.26)** |
| *Country control* | | |
| Neutral | 1.0 | 1.00 |
| Very well controlled | **0.36 (0.18–0.70)** | **2.75 (1.32 5.88)** |
| Somehow controlled | **0.60 (0.41–0.88)** | **1.78 (1.15 2.76)** |
| Not very well controlled | 0.87 (0.58–1.31) | 1.20 (0.76 1.88) |
| Not well controlled at all | 1.42 (0.69–2.92) | 0.94 (0.43 2.04) |
| **Engagement in various activities** | | |
| **Watching Television** | | |
| Same as before | 1.00 | 1.00 |
| Less than before | **0.52 (0.34–0.80)** | **0.47 (0.29 0.75)** |
| More than before | **0.51 (0.35–0.73)** | **0.46 (0.30 0.71)** |
| Prefer not to say/missing data | 1.59 (0.76–3.49) | 1.16 (0.50 2.82) |
| **Time on internet** | | |
| Same as before | 1.00 | 1.00 |
| Less than before | 1.39 (0.80–2.42) | 1.52 (0.82 2.82) |
| More than before | 1.30 (0.88–1.94) | 1.40 (0.91 2.17) |
| Prefer not to say/missing data | 1.19 (0.34–4.36) | 1.56 (0.37 6.94) |
| **Time on social media** | | |
| Same as before | 1.00 | 1.00 |
| Less than before | 1.12 (0.66–1.91) | 1.05 (0.58 1.90) |
| More than before | 0.99 (0.67–1.47) | 1.00 (0.64 1.55) |
| Prefer not to say/missing data | 2.20 (0.65–8.34) | 2.11 (0.54 9.33) |
| **Working for income from home** | | |
| Same as before | 1.00 | 1.00 |
| Less than before | 0.90 (0.60–1.35) | 1.09 (0.70 1.71) |
| More than before | 1.20 (0.84–1.72) | 1.03 (0.68 1.57) |

(*Continued*)

**Table 7.** (Continued)

| Determinants of Interest | Coping | |
|---|---|---|
| | Crude OR (95%CI) | Adjusted OR (95% CI) |
| Prefer not to say/missing data | 0.82 (0.50–1.35) | 0.95 (0.54 1.66) |
| **Performing household chores** | | |
| Same as before | 1.00 | 1.00 |
| Less than before | 0.79 (0.45–1.38) | 0.64 (0.35–1.20) |
| More than before | 0.76 (0.54–1.06) | 0.95 (0.64–1.40) |
| Prefer not to say/missing data | 0.60 (0.27–1.36) | 0.87 (0.36–2.13) |
| **Engaging in sports** | | |
| Same as before | 1.00 | 1.00 |
| Less than before | 1.02 (0.72–1.44) | 1.15 (0.79–1.69) |
| More than before | 0.88 (0.56–1.38) | 0.95 (0.58–1.56) |
| Prefer not to say/missing data | 0.84 (0.47–1.52) | 0.96 (0.50–1.84) |
| **Talking on phone** | | |
| Same as before | 1.00 | 1.00 |
| Less than before | 0.72 (0.41–1.25) | 0.95 (0.51–1.78) |
| More than before | 0.74 (0.53–1.03) | 0.89 (0.61–1.29) |
| Prefer not to say/missing data | 1.25 (0.16–12.26) | 1.27 (0.15–12.62) |
| **Praying** | | |
| Same as before | 1.00 | 1.00 |
| Less than before | **0.58 (0.35–0.97)** | 0.62 (0.36–1.10) |
| More than before | 0.77 (0.56–1.08) | **0.62 (0.43–0.90)** |
| Prefer not to say/missing data | 0.77 (0.30–1.98) | 0.83 (0.30–2.36) |
| **Resting/relaxing** | | |
| Same as before | 1.00 | 1.00 |
| Less than before | 0.79 (0.48–1.31) | 0.84 (0.48–1.49) |
| More than before | 1.28 (0.84–1.95) | 1.28 (0.80–2.06) |
| Prefer not to say | 0.36 (0.00–30.48) | 0.28 (0.00–43.57) |
| **Sleeping** | | |
| Same as before | 1.00 | 1.00 |
| Less than before | **0.55 (0.34–0.89)** | 0.61 (0.36–1.05) |
| More than before | **0.62 (0.40–0.95)** | **0.55 (0.34–0.89)** |
| Prefer not to say/missing data | 0.27 (0.01–10.70) | 0.28 (0.00–16.18) |

that were needed to deal with the "new normal" such as massive closure of schools and public social places, working from home, and wearing of face masks.

Previous studies have shown that the concern of contracting the disease or death is a major source of concern to many people during the pandemic [13, 16]. Consistent with these, we found that majority of the respondents in our study were concerned about their health and that of their families during the pandemic. We also found that nearly one out of ten respondents were extremely concerned about their family finances as a result of the COVID-19. The scale of the pandemic has brought a lot of disruption to the everyday life of many people across the globe. Particularly the lockdown brought much business to a halt. While in the high-income countries, many businesses switched to online, and employees were given the option to work from home, in lower-middle-income countries such as Ghana, the situation was different. There was anecdotal evidence of job losses among those in the private sector and those self-employed. Our finding of most people having extreme concern about family finances

reflects the job losses and job insecurity brought about by the pandemic. Evidence suggests that during the COVID-19 pandemic, persons who had a lack of job security and lack of resources, including financial difficulties were at risk of exposure to stress [16]. This suggests that COVID-19-related stress might be high in our study population.

Many stress coping strategies such as reading, talking to relatives/friends; physical exercise; following a healthy/balanced diet; drinking water to hydrate; following the news; other social media engagements; pursuing hobbies, listening to music, yoga, or gardening; relaxing or doing home chores have been reported to be helpful during crises [16–20]. However, it is argued that the effectiveness of these strategies may be context-specific [14, 21]. A study in Spain documented that where citizens endured longer and stricter lockdown periods there were reductions in risky behaviours with improvement in duration of sleep, and physical activities across genders and age groups [22]. However, Asiamah et al in Ghana found the opposite with an increase in risky health behaviours [23]. This is worth exploring further in our context. In this study, four out of ten respondents reported that they had coped well or better. Furthermore, respondents reported a range of coping strategies employed during the COVID-19, including sleeping, doing house chores, praying, relaxing, and engaging in sports. About 37.6% of participants also indicated that they prayed more during the pandemic and lockdown period and about 29.4% slept more than before. The increased level of prayer could be an indicator of panic and fear among such individuals who were not coping well. As found in studies among participants in Columbia and other countries, religion had a potential impact on coping [24, 25]. This should be further explored in countries like Ghana where religion is a part of the life of the majority of citizens. Many churches in Ghana make use of online services to engage their members in religious activities. The nature and content of such engagement could be harnessed and used as a system for psychosocial support. Sleeping more during such a crisis could be positive or negative. People need to find activities that engage them positively and reduce boredom during the period of lockdown and increased period of staying home. In the absence of such, some resort to excessive sleeping which can be a reflection of boredom or even worse as a potential indicator of depression. This also needs to be explored further. In addition, how people interacted with the media especially for information on the pandemic was found to also predict coping in other studies [3, 16, 26, 27]. In this study, 45.4% and 46.4% of the respondents spent more time on social media and talking on the phone respectively during the outbreak of the COVID-19 pandemic than before. The content of media messages is thus crucial if it will contribute to positive coping strategies and could be explored to offer counseling and other mental health supports [28–30].

The majority of the participants in this study were young adults and lived in urban Ghana. This is not surprising because of the mode of data collection; the younger population are urban dwellers and have access to smartphones in Ghana [23]. Age, sex, and other socio-demographic characteristics are very important factors to coping levels and strategies as reported in other studies [10, 31–33]. A study in Spain found that men and the younger population were worse affected psychologically during the lockdown [34]. Our study suggests that compared to older adolescents, younger adults were less likely to cope with the pandemic while those age 65+ were more likely to cope again in comparison with older adolescents. Context-specific factors such as household arrangements and dependence could account for these differences [14, 21].

Ghana is classified as a lower-middle-income country and from the study, only 2% of participants indicated that they belonged to the high-income economic category. The odds of not coping well was higher in those in the low-income group. The financial security among those of the high-income status understandably will reduce the effect of any economic impact the pandemic had on people. Also, the majority of participants in this study were involved in non-governmental employment. This implies that they are not monthly salary workers and thus

depend on informal sector jobs to earn a living. This has implications for coping as such sectors were more affected by the pandemic; with people who rely on such sources of income suddenly having their income reduced as seen in some studies from Ghana [12, 35].

Another significant finding in this study is the degree to which having a pre-existing condition decreased the likelihood of coping well with the effects of COVID-19 relative to those without a pre-existing condition. Since the beginning of this pandemic, it has been reported and proven that people with these medical conditions like hypertension, asthma, diabetes, and heart diseases among others increased the risk of getting the severe form of COVID-19 disease and its associated higher mortality [36]. Thus, such factors among individuals or as related to their family members or significant others have been found to impact coping negatively as it breeds fear [2]. Such people therefore might require a better assessment to enable them to receive the specific care needed to help them cope well [37].

This study has several strengths. To the best of our knowledge, our study is the largest cross-sectional design into coping and associated factors in sub-Saharan Africa. The design was a population-based cross-sectional study of a section of Ghanaians thus minimizing selection bias. We also used a validated questionnaire which has been used in 10 countries in the Global south. The study allowed participants to complete the questionnaire electronically and that was convenient. However, a number of limitations must also be noted. The study relied on self-reporting and thus, recall bias is a possibility. It can be argued that this study was carried out during the peak of the COVID-19 crisis and it is possible participants could recall their experience vividly. The mode of data collection was completely electronic and this meant that people without access to electronic devices and/or skills to use them were excluded from participating. This could be a source of selection bias and thus affects the generalizability of the findings. In fact, with the global widespread restrictions on COVID-19 and health risks that face to face data collection is associated with, this electronic mode was what had to be used. In addition, the cross-sectional nature of the study precludes any causal relation.

## Conclusions

Overall, this study suggests that the majority of Ghanaians were extremely concerned about their own health and that of a family member during the COVID-19 pandemic. Also, financial security and optimism about the disease, that is having the feeling that life is better during the pandemic, increases one's chances of coping. However, having pre-existing medical conditions decreases the chances of coping. Furthermore, praying more and sleeping more during the pandemic than before reduces one's chances of coping. These findings should guide public health policy and mental health intervention during the ongoing COVID-19 pandemic as well as a future public health crisis.

## Supporting information

**S1 Appendix. Questionnaire.**
(PDF)

**S1 Dataset.**
(XLSX)

## Acknowledgments

We acknowledge the contribution of the rest of the global team for this multinational research. The Director of the Directorate of Research, Innovation, and Consultancy, the University of Cape Coast for his support and director during this project.

## Author Contributions

**Conceptualization:** Dorcas Obiri-Yeboah, Stefan Jansen, David Teye Doku, Frederick Ato Armah.

**Data curation:** Samuel Iddi, Dorcas Obiri-Yeboah, Stefan Jansen, David Teye Doku, Frederick Ato Armah.

**Formal analysis:** Samuel Iddi, Dorcas Obiri-Yeboah, Reginald Quansah, Stefan Jansen.

**Investigation:** Dorcas Obiri-Yeboah, Irene Korkoi Aboh, Samuel Asiedu Owusu, Nancy Innocentia Ebu Enyan, Ruby Victoria Kodom, Epaphrodite Nsabimana, Stefan Jansen, Benard Ekumah, Sheila A. Boamah, Godfred Odei Boateng, David Teye Doku, Frederick Ato Armah.

**Methodology:** Samuel Asiedu Owusu, Nancy Innocentia Ebu Enyan, Ruby Victoria Kodom, Epaphrodite Nsabimana, Stefan Jansen, Benard Ekumah, Sheila A. Boamah, Godfred Odei Boateng, Frederick Ato Armah.

**Project administration:** Stefan Jansen, Frederick Ato Armah.

**Supervision:** Dorcas Obiri-Yeboah, Stefan Jansen, David Teye Doku, Frederick Ato Armah.

**Validation:** Samuel Iddi, Reginald Quansah, Epaphrodite Nsabimana, Godfred Odei Boateng, Frederick Ato Armah.

**Writing – original draft:** Samuel Iddi, Dorcas Obiri-Yeboah, Irene Korkoi Aboh, Reginald Quansah, David Teye Doku.

**Writing – review & editing:** Samuel Iddi, Dorcas Obiri-Yeboah, Irene Korkoi Aboh, Samuel Asiedu Owusu, Nancy Innocentia Ebu Enyan, Ruby Victoria Kodom, Epaphrodite Nsabimana, Stefan Jansen, Benard Ekumah, Sheila A. Boamah, Godfred Odei Boateng, David Teye Doku, Frederick Ato Armah.

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
