## [Decision Letter · Decision Letter 0]

19 Apr 2021

PONE-D-21-03531

Coping Strategies Adapted by Ghanaians during the COVID-19 Crisis and Lockdown; a Population-based Study

PLOS ONE

Dear Dr. Iddi,

Thank you for submitting your manuscript to PLOS ONE. After careful consideration, we feel that it has merit but does not fully meet PLOS ONE’s publication criteria as it currently stands. Therefore, we invite you to submit a revised version of the manuscript that addresses the points raised during the review process.

An expert in this field and myself have carefully reviewed your submission. Both of us believe that your work has some merits and is publishable should you do some revisions. Please take the comments from the reviewer seriously when you prepare a revision.

We look forward to receiving your revised manuscript.

Kind regards,

Chung-Ying Lin

Academic Editor

PLOS ONE

Journal Requirements:

Please provide additional details regarding participant consent. In the ethics statement in the Methods and online submission information, please ensure that you have specified what type you obtained (for instance, written or verbal, and if verbal, how it was documented and witnessed). If your study included minors, state whether you obtained consent from parents or guardians. If the need for consent was waived by the ethics committee, please include this information.

During our internal checks, the in-house editorial staff noted that you conducted research or obtained samples in another country. Please check the relevant national regulations and laws applying to foreign researchers and state whether you obtained the required permits and approvals. Please address this in your ethics statement in both the manuscript and submission information."""

During the internal evaluation of the manuscript we have noted that the current study is a part of a a multi-country online cross-sectional survey on Personal and Family Coping with COVID-19. Please provide a citation for this study.

Please include additional information regarding the survey or questionnaire used in the study and ensure that you have provided sufficient details that others could replicate the analyses. For instance, if you developed a questionnaire as part of this study and it is not under a copyright more restrictive than CC-BY, please include a copy, in both the original language and English, as Supporting Information. Furthermore, please provide additional information regarding the questionnaire development and validation process, including the theories or frameworks which were employed.

Please include as Supporting File a list of all participating countries.

We note that you have indicated that data from this study are available upon request. PLOS only allows data to be available upon request if there are legal or ethical restrictions on sharing data publicly. For information on unacceptable data access restrictions, please see http://journals.plos.org/plosone/s/data-availability#loc-unacceptable-data-access-restrictions.

5a) If there are ethical or legal restrictions on sharing a de-identified data set, please explain them in detail (e.g., data contain potentially identifying or sensitive patient information) and who has imposed them (e.g., an ethics committee). Please also provide contact information for a data access committee, ethics committee, or other institutional body to which data requests may be sent.

5b) If there are no restrictions, please upload the minimal anonymized data set necessary to replicate your study findings as either Supporting Information files or to a stable, public repository and provide us with the relevant URLs, DOIs, or accession numbers. Please see http://www.bmj.com/content/340/bmj.c181.long for guidelines on how to de-identify and prepare clinical data for publication. For a list of acceptable repositories, please see http://journals.plos.org/plosone/s/data-availability#loc-recommended-repositories.

Reviewers' comments:

Reviewer's Responses to Questions

**Comments to the Author**

1. Is the manuscript technically sound, and do the data support the conclusions?

Reviewer #1: Yes

2. Has the statistical analysis been performed appropriately and rigorously? 

Reviewer #1: Yes

3. Have the authors made all data underlying the findings in their manuscript fully available?

Reviewer #1: No

4. Is the manuscript presented in an intelligible fashion and written in standard English?

Reviewer #1: Yes

5. Review Comments to the Author

Reviewer #1: Abstract:

1. Line 30: Authors may revise the sentence to “This study was..” by adding “study”

Materials and Methods

2. Line 82-83: “Participants were recruited from different social media platforms and personal contact via phone calls.” How can authors re-assure readers that a participant didn't complete the questionnaire more than once? Any safeguards?

3. Line 86: Is it “Cohen’s d” or “Cohen’s dz?

4. Line 95: “The online questionnaire is composed of different psychometrically validated instruments”. Can authors report the reliability and validity values for these scales among Ghanaian samples?

Results

5. Lines 179-190: Especially the first sentence (ordered logistic regression are shown in Table 6) seems to be misleading as it did not reflect in Table 6. Hence, authors may re-look at it and revise it appropriately.

6. Lines 193-206: Authors seem to report data (figures) that cannot be verified from its respective Table (Table 7). Also, there were typos (e.g., line 201).

Discussion

7. Line 229: Does Ghana belong to low-income country?

8. Line 235-238: Please cite some studies to support the sentence.

9. Line 244-245: “A total…” can authors use percentages in the sentence instead of absolute figures?

10. Lines 245-246: “These were found to be associated with decreased coping.” What are “these”?

11. Lines 270-271: “…will reduce the impact of any economic impact the pandemic had on people.” Can authors replace of the impacts with another word?

12. Lines 276-277: “…relative to those with a pre-existing condition.” Is it “without” rather?

13. There are few grammatical and typos. Hence, authors may have to proof-read and revise appropriately.

6. PLOS authors have the option to publish the peer review history of their article (what does this mean?). If published, this will include your full peer review and any attached files.

Reviewer #1: No

---

## [Author Response · Author response to Decision Letter 0]

20 May 2021

Dear Reviewer,

We are grateful for the nice appraisal of the manuscript, and for your valuable comments. 

We have considered all the comments and suggestions, and have provided a point-by-point response to each comments. 

Both the ‘revised manuscript with track changes’ and ‘the unmarked version of the manuscript’ are uploaded as separate files. 

Samuel

---

## [Decision Letter · Decision Letter 1]

14 Jun 2021

Coping Strategies Adapted by Ghanaians during the COVID-19 Crisis and Lockdown; a Population-based Study

PONE-D-21-03531R1

Dear Dr. Iddi,

We’re pleased to inform you that your manuscript has been judged scientifically suitable for publication and will be formally accepted for publication once it meets all outstanding technical requirements.

Kind regards,

Chung-Ying Lin

Academic Editor

PLOS ONE

Additional Editor Comments (optional):

Reviewers' comments:

Reviewer's Responses to Questions

**Comments to the Author**

1. If the authors have adequately addressed your comments raised in a previous round of review and you feel that this manuscript is now acceptable for publication, you may indicate that here to bypass the “Comments to the Author” section, enter your conflict of interest statement in the “Confidential to Editor” section, and submit your "Accept" recommendation.

Reviewer #1: All comments have been addressed

2. Is the manuscript technically sound, and do the data support the conclusions?

Reviewer #1: Yes

3. Has the statistical analysis been performed appropriately and rigorously? 

Reviewer #1: Yes

4. Have the authors made all data underlying the findings in their manuscript fully available?

Reviewer #1: Yes

5. Is the manuscript presented in an intelligible fashion and written in standard English?

Reviewer #1: Yes

6. Review Comments to the Author

Reviewer #1: The authors have revised their manuscript satisfactorily. Hence, I recommend that the study, "Coping Strategies Adapted by Ghanaians during the COVID-19 Crisis and Lockdown; a Population-based Study", be published.

7. PLOS authors have the option to publish the peer review history of their article (what does this mean?). If published, this will include your full peer review and any attached files.

Reviewer #1: No

---

## [Editor Report · Acceptance letter]

17 Jun 2021

PONE-D-21-03531R1 

Coping Strategies Adapted by Ghanaians during the COVID-19 Crisis and Lockdown; a Population-based Study 

Dear Dr. Iddi:

I'm pleased to inform you that your manuscript has been deemed suitable for publication in PLOS ONE. Congratulations! Your manuscript is now with our production department. 

Kind regards, 

on behalf of

Dr. Chung-Ying Lin 

Academic Editor

PLOS ONE